# A lexical decision task for rapid estimation of crystalized vocabulary knowledge in Thai

Graham Pluck[1,2,3], Alexis Sirisomboonwong[1], Smriti Sitani[1], Carl Piaf[4], Suphasiree Chantavarin[1,5]*

1 Faculty of Psychology, Chulalongkorn University, Bangkok, Thailand, 2 Institute of Neurosciences, Universidad San Francisco de Quito, Cumbayá, Ecuador, 3 Department of Psychology, College of Human Sciences and Education, KIMEP University, Almaty, Kazakhstan, 4 Independent Researcher, Bangkok, Thailand, 5 COgnition Audition and Language (CoALa) Research Group, Faculty of Medicine, Chulalongkorn University, Bangkok, Thailand

* suphasiree.c@chula.ac.th

## Abstract

A useful distinction within cognitive and brain sciences is that between fluid and crystallized ability. Although fluid ability is widely studied, crystalized ability, which draws on acquired, declarative semantic knowledge and the mental lexicon, has been less well studied. Partly this is due to the culture and language specificity of assessment methods. We developed and assessed the psychometric properties of a simple 42-item lexical decision task that could be used with Thai speakers to assess the breadth of their crystalized vocabulary knowledge. A large sample of responses from 662 Thai-speaking participants, collected online, was used to refine the scale through exploratory factor analysis, and establish its internal consistency. A smaller sample of 90 participants was interviewed to establish validity of the task as a measure of Thai vocabulary. Large positive correlations (i.e., > .3) were found between the Thai Lexical Decision Task and measures of verbal fluency, particularly in the first 30 seconds of responding, and with other measures of Thai language skill. Temporal stability of the scale was assessed in a subsample of 27 participants. This confirmed that the Thai Lexical Decision Task has little obvious practice effect and excellent test-retest reliability. We also observed the expected positive associations with age, educational level, and self-reported proficiency in the language, supporting the ability of the task to measure acquired, crystalized knowledge. This new task could be used to estimate cognitive ability in Thai adults and shows potential as a measure of premorbid cognitive function for use in neuropsychological assessments.

## Introduction

A basic distinction, widely accepted in the various cognitive and neural sciences, is between active, 'fluid' cognitive-control ability, and 'crystallized' ability using stored knowledge, particularly related to semantic memory. The most well-known

**Data availability statement:** The Thai Lexical Decision Task is free to use in research without cost, and can be downloaded from: https://gpluck.co.uk/Tests/ The data files can be found on this repository: https://doi.org/10.23668/psycharchives.16526.

**Funding:** This project is supported by Ratchadaphiseksomphot Fund, Chulalongkorn University (Cognition, Audition and Language (CoALa) Research Group).

**Competing interests:** The authors declare that no competing interests exist.

version is Cattel's distinction between fluid and crystallized intelligence [1], but it is an integral component to many other approaches. One notable example is Baddeley's multi-component working memory model [2], and another comes from intelligence testing, in which the distinction between 'performance' and 'verbal' IQ is equivalent to fluid and crystalized [3]. The fluid/crystalized contrast is useful, as the two factors have very different features, particularly in their relationships with formal education [4], developmental trajectories [5], and neurological disorders [6,7]. These are due to the learnt nature of crystalized ability, which increases with age and educational experience, and is quite resilient in the presence of normal and pathological cognitive ageing [5]. Nevertheless, few standardized tests of crystallized ability are available, even in English, as most psychometric studies have focused on fluid abilities such as non-verbal intelligence, working memory, and attention.

Fluid ability is characterized by active processing for immediate behavior, such as spatial navigation and decision making; it is very closely associated with complex problem solving [8] and executive functions [9] and may be independent of language skill [10]. In contrast, crystalized ability is closely linked with declarative semantic knowledge, vocabulary, and language skill [11]. However, crystalized ability is more than stored knowledge; it is also linked to intelligence and includes flexible thinking, particularly creativity [12]. It is likely that both executive 'fluid' and semantic 'crystalized' control both contribute to performance in a range of cognitive tasks, such as cognitive estimation [13] and verbal fluency [14], but at the neural level, they are dissociable [15].

Crystalized ability encompasses much about declarative memory, particularly semantic memory, including various dimensions of general knowledge [16]. Declarative semantic memory also includes the mental lexicon, the set of links between conceptual meaning, and, at its most basic form, the phonemes that indicate those meanings [17]. The different aspects of factual knowledge and the lexicon are very closely associated [11].

### Methods of assessment of crystalized ability

Assessment of a person's lexicon via their vocabulary knowledge is the most common method of measuring their crystalized ability. This is because within linguistic cultures, there is a common consensus over the meanings of a finite number of words. By measuring performance on a task of the knowledge of a sample of words, one has an idea of the breadth of the mental lexicon on an individual.

One very common method that psychologists use to achieve this is through asking for definitions of words. An example would be a participant being asked "what does the word 'nationality' mean?", and the researcher/clinician scoring the response for accuracy in terms of commonly understood meanings of the word. Such tests are included in many intelligence test batteries, such as the Weschler family of tests [3]. These tests, though, as acts of communication between the participant and researcher/clinician, also likely assess non-lexical processes, such as fluency, mentalizing skill, and social communication adeptness. A method that reduces the social

and subjective aspects of performance is to use multiple-choice assessments of vocabulary, based on matching words to phrases that are synonyms [18].

An alternative is to use lexical decision tasks. In such tasks, real words are presented with pseudowords (i.e., words that appear to be legitimate words in the language but are not actually used in the language). One method, widely-used for language assessment in educational contexts, is to present a mixed array of words and pseudowords, and the test-taker must choose which they know, and which they do not. Such recognition yes/no tests have been widely-used for student placement within a language education context, because, despite their simplicity, achievement scores correlate very highly with scores on more comprehensive and lengthy language-skill assessments [19]. They also appear to have favorable psychometric properties when compared to multiple-choice vocabulary assessment methods [20]. The success of those lexical-decision studies led to the development of LexTALE, a simple lexical-decision based method for assessing vocabulary acquisition in a range of second-language contexts [21]. In addition to the English version of LexTALE, there are versions available for several different languages including French [22], German, and Dutch [21]. Although designed for use in second-language assessment, these tests often do not have ceiling effects in first-language speakers, suggesting they may have use in other contexts. Lexical decision is also one component of the DIALANG system, also used to assess competence of teenagers and adults learning a range of second languages [23].

The first such standardized test of this type for psychological research purposes may have been the German-language Mehrfachwahl-Wortschatz-Intelligenztest B (multiple-choice vocabulary intelligence test B; [24]). In that test, German words must be identified from sets composed of one real word plus four pseudowords. In the English language, the Spot-the-Word test [25,26], originally part of the commercially sold Speed and Capacity of Language Processing Test [27], is the most applied lexical decision task for cognitive assessments. In that test, pairs of words, such as 'Puma - Laptess', are shown to the participant and they must choose the real word from each pair. In this case 'Puma' would be the correct choice, as 'Laptess' is a pseudoword.

Lexical decision has several benefits as a testing method. On the part of the participant, it does not require articulation, and there is an option to guess, thus avoiding potential embarrassment from being asked to define words one does not understand. On the part of the researcher/clinician, scoring does not require listening or reading, and so can be relatively automated, allowing for online or group test administration. It is also objectively scored, eliminating any variation in scores due to imperfect interrater reliability.

A further benefit is that when lexical knowledge is to be used as a measure of crystalized ability, it is useful to minimize any fluid processing aspects of task performance. The latter of these requires some level of top-down executive control, shown by its sensitivity to cognitive load [28], while recognition has a substantial element of implicit memory, and is generally unaffected by cognitive load [29]. This is underlined by second language acquisition studies that show that vocabulary recognition is generally easier than recall [30], and is a better predictor of reading comprehension because performance of the task requires a narrower range of cognitive processes, offering a purer performance measure [31]. From a neuropsychological perspective, a reason for this dissociation is that lexical decision tasks can be performed without access to conceptual knowledge; to respond correctly it is enough to have a feeling that a word is real, that one has seen that word before, even if one is unable to locate the meaning of the word [25].

Nevertheless, current models of lexical decision task performance suggest that multiple sources of information are used to reach a decision, and this includes how 'word-like' the pseudowords seem [32]. It is therefore important to minimize such information, by providing only convincing pseudowords.

## Availability of validated lexical decision tasks

Few standardized lexical decision tasks are available for clinical or research use in psychology. One reason for this is that they are, by their nature, language and culture specific. The Spot-the-Word Test was developed in the UK and uses British English words, indeed some words (e.g., 'octaroon'), though listed in British dictionaries, are not listed in dictionaries from

the USA, casting some doubt on its validity within that country. And of course, such tests are not valid for people who speak languages other than English. Consequently, localized versions need to be developed.

Lexical decision tasks very similar to the Spot-the-Word Test have been developed for use in several languages, including Spanish [33], and Brazilian Portuguese [34]. There is also a version in Swedish that uses a different format but is still a lexical-decision task [35]. However, no standardized test of crystalized ability exists for clinical or psychological research use in the Thai language. Development of such a task would assist future research in psychology and the behavioral sciences within the country. Here, we describe studies of the validity and reliability of a Thai Lexical Decision Task. The target population is adults who speak Thai as their main language. Study 1 reports the development of the task and the use of factor analysis with a large online sample to refine the scale and to establish its internal consistency. Study 2 aimed to establish the validity of the task by investigating performance correlations between the lexical decision task and measures of Thai verbal fluency. Finally, Study 3 aimed to assess the test-retest reliability of the scale in a subsample of participants.

## Study 1: Task development and factor analysis with an online sample

### Method

**Aims, study design and hypotheses.** The aim of this study was to pilot an initial set of words and pseudowords to explore item difficulty and scale unidimensionality with a large sample of participants. An online study was conducted with an initial version of the Thai Lexical Decision Task. We also hypothesized that as a measure of acquired crystalized knowledge, scores on the task would be positively associated with age, education level, and self-rated language proficiency.

**Participants.** A total of 992 participants were recruited online (i.e., completed at least the consent stage) from the Chulalongkorn University community. Although these were recruited through convenience sampling, this approach is useful as it allows recruitment from a set geographical area, and convenience samples can be appropriate when used to pilot assessments and generate material for further studies [36], as is being done in the current study. The inclusion criteria to participate were that an individual was aged 18 or above, a Thai national, and was a fluent native Thai speaker (exposed to Thai language before the age of 5). The recruitment period for the whole study reported in this paper was between March 13, 2023 to March 8, 2024.

The mean age of the whole sample was 32.6 years ($SD = 7.1$, range = 18–79), and most identified as female, 641/992 (65%). However, many cases were excluded during data quality processing, the actual analyzed sample is thus described later.

**Development of the initial version of the Thai Lexical Decision Task.** To develop a large set of target words for the Thai Lexical Decision Task, we initially selected 100 real Thai words. All words were selected to be concrete concepts that were imageable; the relevance of this feature is described later. Since participant performance is evaluated based on their accuracy rather than their response times, the task had to be designed to be moderately difficult. Thus, we purposely selected some words that had low frequency of use (e.g., ชลมารค [royal barge], ผอบ [casket], ปฏัก [goad stick]) so that these words will not be easily distinguishable from nonwords. We also included colloquial words that were relatively common and familiar in everyday conversation (e.g., จราจร [traffic], ศีรษะ [head], ปีศาจ [ghost]). In terms of etymology and register, most of the selected words were literary Thai lexemes of Sanskrit/Pali origin, as well as a few royal vocabulary, archaic poetic words, and formal words (see Table 2). These items were intentionally skewed towards orthographically complex Indic-derived words drawn primarily from non-colloquial registers in order to avoid ceiling effects and ensure sufficient variability in accuracy and response times across participants. The frequency of the selected words within the Thai language varied from < 1 to 70.2 per million ($M = 5.9$, $SD = 13.7$), estimated using the Thai web 2018 (thTenTen18) corpus from the Sketch Engine website (https://www.sketchengine.eu/). The word length ranged from 2–7 characters ($M = 4.1$, $SD = 1.2$), defined as the number of consonant and vowel characters excluding the vowels and tone marks located above or below those characters. The number of syllables ranged from 1–4 ($M = 2.2$, $SD = 0.7$). Research

assistants looked up each word in search engines and dictionaries to verify that each of the words had meaning in the Thai language.

A list of 100 pseudowords was created to be paired with the real words. These were produced by a Thai doctoral level psycholinguist, without the use of AI. The pseudowords were designed to look and sound like plausible Thai words without violating spelling conventions; all were pronounceable and orthotactically legal, but did not correspond to any attested Thai words. Many of the pseudowords were formed by combining syllables or orthographic segments from existing Thai words, e.g., จุติ and หรรษ์ are attested words in Thai, but their combination จุติหรรษ์ is not a word in Thai. Some of these were created using grapheme segments that resemble real morphemes in Thai but contain consonant or vowel variants, e.g., จรรญาพรม, ทัศภาติ, กรรโดษฐ์, ภิเมษฐา, ศยามาร. Others were short, monosyllabic pseudowords which were generated to preserve typical Thai syllable structure without directly recombining identifiable morphemes, e.g., เฌบง, กรวก, ขวัง, ตจุลย์, ครวด. The pseudowords contained a similar set of letters as the real words, containing high-frequency letters (e.g., ก, ช, ส) as well as lower-frequency letters (e.g., ณ, ฐ, ษ), combined in different ways. The length of the pseudowords ranged from 2–8 characters ($M = 4.2$, $SD = 1.2$), and the approximate number of syllables ranged from 1–4 ($M = 2.0$, $SD = 0.8$). Research assistants verified through online searches that the pseudowords did not have a conventional meaning in Thai language.

The real words and pseudowords were randomly paired to create the 100 test items. Since the random pairing resulted in some very easy items (e.g., a high-frequency real word was paired with a pseudoword that contained low-frequency letters, thus providing a cue that it was a pseudoword), according to pilot data, we manually switched some pairings to avoid creating real word – pseudoword pairs that were too easy. For task performance, the variable of interest is accuracy, that is, whether the correct word in each pair is selected or not.

**Procedure.**  The procedure, including written consent taking, had approval of the local research ethics committee (Study Title No. 660010; Approved by the Research Ethics Review Committee at Chulalongkorn University on March 9, 2023) and the research was performed in compliance with all relevant laws and institutional guidelines. The initial version of the Lexical Decision Task was placed online using Qualtrics. The online task with 100 real word – pseudoword pairs was promoted on a social media account of the Faculty of Psychology at Chulalongkorn University, Bangkok. When interested individuals clicked on the link, they were taken to the Qualtrics site and the research was explained to them in text form, at which point they provided consent to continue by checking a box. Initial additional questions in the survey asked about the participants' (a) age, (b) gender, (c) highest educational attainment, (d) exposure to Thai language before the age of 5 (options being 'no exposure', 'Thai only', and 'Thai and other languages'), (e) and a five-point self-rating of proficiency in Thai language from 'very low (1)' to 'very high (5)'.

Next, participants read the instructions specifying that they will see pairs of words comprising of a real word and a pseudoword that does not have a meaning in Thai language, and that their task was to select the word in each pair that was a real word. An example item and correct response was provided. The main task was presented with pairs of real and pseudowords displayed in the standard default font in a legible 17 pt size, with all 100 items appearing on the same page on the screen. The locations of the real words and pseudowords were randomized so that the real words/ pseudowords did not always appear consistently on either the left or right side. At the end of the task, participants were told how many they had answered correctly. This feedback element appeared to make the task challenging and the initial social media link was shared many times, resulting in the task being attempted 989 times over a two-day period. At this point the task was taken offline.

## Results and discussion of the online study

**Data processing.**  The initial data set from Qualtrics contained 992 completed consents, and 989 participants proceeded to respond to questions. We performed data cleaning to improve the quality of the dataset by removing cases with incomplete data. We also removed cases completed while outside of Thailand, based on IP-addresses (as these may have included non-native Thai speakers), and subsequent attempts from the same IP address (as these may have

indicated people completing the task more than once). Some attempts had been completed in an unfeasibly short period of time, a behavior associated with low data quality [37]. Other attempts had taken an unreasonably long time to complete and may indicate participants researching the words before answering, such as checking in dictionaries. Cut offs of < 180 seconds and > 900 seconds for task completion were used as reasonable criteria for removing temporally invalid records. Of the retained cases, the mean completion time was just under seven minutes (412 seconds), equivalent to completing the task by spending less than 4 seconds on each choice made. Using combined geographical filtering and IP address is an effective method for reducing fraudulent data produced by bots or multiple attempts by the same individual [38].

**Demographics of the retained sample.** Of the retained sample of 662 participants, the mean age was 32.6 years ($SD = 6.8$). Two-thirds (66%) of the sample identified as female, 154/662 as male (23%) and the remainder (11%) identified as 'other'. Self-reported education level was generally high, likely a consequence of the online task being promoted from a university social media account. Only 28/662 participants (4%) reported no formal education beyond high school. The majority, 353/662 (53%) reported undergraduate education, and 278/662 (42%) reported graduate level education. The sample, therefore, overrepresents the most highly educated section of Thai society. All participants reported exposure to Thai language before the age of five, but 232 (35%) of them indicated that they were exposed to both Thai and other languages during that period. Demographic variables are summarized in Table 1.

**Analysis of individual items and factor analysis.** In the next stage of analysis, the accuracy of performance from the 662 participants was examined on an item-by-item basis. One item (#53) was found to have very low accuracy, and on checking, despite being included in the initial task, the 'real' word was not found in any Thai dictionaries. Consequently, responses to that item were removed from the data set. Several other items appeared to produce very little variance in response, with correct responses from the majority of participants. Insufficient item difficulty is potential challenge to validity and should be addressed in test development [39]. We examined this in both the full sample of 662 participants, and the subsample of 28 participants with low education levels (high school or lower). Items were removed from the data set if accuracy in the full sample was > 97.5% (i.e., more than 645 of the 662 participants answered correctly). However, items were retained if scored as < 94% correct in the lower-education subsample (i.e., at least two of the 28 participants made an error). This allowed us to retain items that may be challenging to less educated participants. This resulted in the removal of 31 further items. This pruning was necessary as items with very low variance would not be psychometrically useful, in that low variance within items tends to reduce scale reliability [40], and also would not be appropriate for factor analyses. There were also 6 items that were answered at < 50% accuracy and were removed at this stage. As the chance guessing rate is 50%, they may indicate items in which the pseudoword was perceived as more word-like than the real word. As such, they reflect the interpretation of the pseudoword more than the target word and lack face validity as a test of vocabulary. Together, this totaled removal of 38 of the original items, leaving 62 items with a reasonable level of variance.

Next, the factor structure of the data set from the 662 participants on the 62 lexical-decision items was examined. Because the responses to individual items by individual participants have Boolean values expressing whether they were answered correctly or not (1 = true, 0 = false), Pearson correlations cannot be used in exploratory factor analysis. Instead,

**Table 1. Summary of demographic characteristics of the study participants.**

|  | Study 1 (online) | Study 2 (interview) | Study 3 (interview) |
|---|---|---|---|
| n | 662 | 90 | 27 |
| Mean Age (+SD) | 32.6 (6.8) | 36 (16.7) | 37.5 (18.3) |
| % Female | 66 | 69 | 67 |
| % Completed Graduate Education | 42 | 17 | 22 |
| % with Childhood Multilingual Exposure | 35 | 46 | 44 |

polychoric correlations are recommended [41]. The use of polychoric correlations for the Boolean data of observed variables assumes that there is a continuous and normally distributed scale underlying each of the observed variables. This assumption is reasonable in the sense that the vocabulary knowledge of individuals is widely conceptualized as existing on a continuum (e.g., [42]). Polychoric correlation coefficients range from 0 to 1, in which the values closer to 0 indicate weaker association and those closer to 1 indicate stronger association.

Next, a 62 x 62 matrix of the polychoric correlation values was submitted to exploratory factor analysis. The estimation method used was the iterated principal factor method with the prior communality for each variable set to the squared multiple correlation with all other variables. In addition, the number of factors to be extracted was set to one because all the items are intended to measure only the breadth of the mental lexicon. The scree plot of eigenvalues confirmed the specification of a single factor. Finally, it is thought that loadings < .3 do not meet the minimal level for inclusion within a factor [43]. With that threshold, 20 additional items were removed from the scale. Consequently, the final Thai Lexical Decision Task had 42 remaining items. Those items, along with their factor loadings, are shown in Table 2. Twenty of these 42 words were relatively high-frequency words (defined as 1 word per million and above; $M = 4.82$, $SD = 6.04$, range = 1.01–23.3), and the average number of characters in this set was 3.35 ($SD = 0.88$, range = 2–5 characters). The remaining words were relatively low-frequency words (defined as lower than 1 word per million; $M = 0.19$, $SD = 0.27$, range = 0–0.99), and the average number of characters in this set was 4.64 ($SD = 1.09$, range 3–7 characters).

**Final version of the Thai Lexical Decision Task.** Those 42 items were used to calculate a Thai Lexical Decision Task total score by summation. Psychometric properties of that total score are shown in Table 3. The internal consistency of the whole scale was estimated with Kuder-Richardson 20 (KR-20) which is equivalent to Cronbach's alpha, but used with dichotomous scores [44]. This was found to indicate 'good' internal consistency. Skewness and kurtosis are not included in the table as it also contains data from a separate, smaller sample (described in Study 2), and ways to interpret non-normality vary by sample sizes [45]. However, in the current (online) sample the absolute skewness value of −2.24 suggests substantial negative skewing of the data distribution, which is also shown by the central tendency (mean = 38, median = 39) being close to the maximum (42). This likely reflects the high education level of the sample. The distribution was also highly leptokurtic, with an absolute value of 10.1, which would also indicate non-normal distribution, probably also linked to the highly skewed education level of the sample. Given the highly non-normal distribution, non-parametric correlations were used to examine associations with demographic factors. Gender was dichotomized (male or female). Thai Lexical Decision Task total scores had moderate associations with education level and self-reported Thai proficiency, qualitatively 'typical' positive associations, and a qualitatively 'typical' to 'large' positive association with age. Qualitative descriptions of correlation coefficients used in this paper are based on a standardized interpretation [46,47]: $r > .1$ = 'small', $r > .2$ = 'typical', and $r > .3$ = 'large'). The scatterplots for those associations are shown in Fig 1. There was a very small, but significant, zero-order correlation of task performance with gender, indicating slightly better performance by female participants. Of the 20 words that we classified as high frequency, the mean recognition score was 18.3 (SD = 2.0, 92% correct), which is very similar to the score for the 22 low frequency words which had a mean recognition score of 20.01 (SD = 2.2, 91% correct). That small difference was not statistically significant ($t$(661) = 1.90, $p = .058$ (two-sided), $d = 0.074$).

**Summary of the online study.** To summarize, the findings from the online study were used to prune an original set of 100 items to a set likely to perform well as a single scale of crystalized vocabulary knowledge. Although the sample used was large, it also contained an overrepresentation of highly educated participants. Nevertheless, we were able to select a set of 42 items which were challenging to many participants and loaded onto a single factor, assumed to represent vocabulary knowledge. One question raised is whether this scale will perform well when applied to a more educationally diverse sample; that issue is dealt with in the next studies. Although the devised 42-item scale has obvious face validity as a measure of vocabulary knowledge in Thai, its psychometric validity remains to be demonstrated formally. Thus, a

 

**Table 2. Items included in the final version of the Thai Lexical Decision Task based on analyses of the online study (*n*=662).**

| Item Number | Word Pair | English translation of real word | Etymology of real word | Register of real word | Accuracy | Factor Loading |
|---|---|---|---|---|---|---|
| 1 | **กบาล** — เกือน | Head, skull | Sanskrit (*kapāla*) | Informal | 98.8 | .78 |
| 2 | คเมช — **จวัก** | Ladle | Thai | Colloquial | 97.9 | .49 |
| 3 | **โอษฐ์** — ทัศภาติ | Lips | Sanskrit (*oṣṭha*) | Royal | 97.6 | .93 |
| 4 | ทรรษา — **นาคา** | Snake, serpent deity | Sanskrit (*nāga*) | Literary | 98.9 | .57 |
| 5 | **ทิชากร** — ถานร | Bird | Sanskrit | Poetic | 98.2 | .81 |
| 6 | ขจัก — **ศกุนี** | Female bird | Sanskrit (*śakunī*) | Literary | 71.9 | .39 |
| 7 | มักษา — **นริศ** | King | Sanskrit (*nareśa*) | Literary | 92.7 | .64 |
| 8 | **ชลมารค** — เฌบง | Waterway | Sanskrit | Literary | 94.1 | .52 |
| 9 | เนศไรย์ — **กรกฎ** | Crab | Sanskrit (*karkaṭa*) | Formal | 97.1 | .55 |
| 10 | เกลบ — **ผอบ** | Casket | Thai | Formal | 93.2 | .63 |
| 11 | **อรรณพ** — ศุกานะ | Sea, ocean | Sanskrit | Poetic | 95.5 | .47 |
| 12 | **พาชี** — กรวก | Horse | Sanskrit | Literary | 88.8 | .42 |
| 13 | **เขฬะ** — กลาน | Saliva | Sanskrit | Literary | 91.5 | .66 |
| 14 | **ทิพา** — จรรญาพรม | Day | Sanskrit (*divā*) | Literary | 89.3 | .64 |
| 15 | ปฉาน — **มุสิก** | Rat | Sanskrit (*mūṣika*) | Literary | 96.7 | .64 |
| 16 | **ทูต** — ถนาน | Ambassador | Sanskrit (*dūta*) | Formal | 97.1 | .42 |
| 17 | ชลวาร — **รวิ** | Sun | Sanskrit (*ravi*) | Poetic | 63.6 | .51 |
| 18 | ธุรพี — **หิรัญ** | Silver | Sanskrit (*hiraṇya*) | Poetic | 95.9 | .64 |
| 19 | **นขลิขิต** — รัคนี | Parentheses | Sanskrit | Formal | 92.1 | .48 |
| 20 | โทรจักร — **นาคี** | Snake | Sanskrit | Poetic | 97.1 | .42 |
| 21 | **สิว** — ชะมอน | Chisel | Thai | Colloquial | 97.6 | .51 |
| 22 | ศาตรี — **นลาฏ** | Forehead | Sanskrit (*lalāṭa*) | Literary | 76.3 | .46 |
| 23 | **บัณเฑาะว์** — เสษณี | Type of drum | Sanskrit | Literary | 93.4 | .64 |
| 24 | **มหิงส์** — จุติหรรษ์ | Buffalo | Sanskrit (*mahiṣa*) | Literary | 93.7 | .46 |
| 25 | พะนาจ — **ดารกะ** | Stars | Sanskrit (*tārakā*) | Literary | 73.0 | .35 |
| 26 | **เหม** — บูรหิต | Gold | Sanskrit (*hemā*) | Literary | 85.2 | .61 |
| 27 | นาช — **จุรี** | Sword | Khmer | Literary | 95.3 | .48 |
| 28 | มลจรรย์ — **ภุชงค์** | Snake, serpent | Sanskrit (*phujaṅga*) | Poetic | 97.1 | .61 |
| 29 | **ปรัศนี** — สาวกา | Question mark | Sanskrit (*praśna*) | Formal | 88.7 | .40 |
| 30 | สรรไตร — **ปโฏก** | Goad stick | Sanskrit (*pratoda*) | Formal | 86.4 | .51 |
| 31 | **กรกช** — โฉตน | Lotus | Sanskrit | Poetic | 95.3 | .66 |
| 32 | **สิตางศุ์** — กรรโฑษฐ์ | Moon | Sanskrit (*Sitāṃśu*) | Poetic | 92.7 | .42 |
| 33 | เหริง — **มัชฌิมา** | Middle finger | Sanskrit (*madhyama*) | Poetic | 97.1 | .73 |
| 34 | พานี — **เฌอ** | Tree | Khmer | Literary | 83.4 | .33 |
| 35 | **มกุฏ** — ปราล | Crown | Sanskrit (*mukuṭa*) | Royal | 90.0 | .58 |
| 36 | **กระตัว** — ตจุลย์ | Cockatoo | Thai | Colloquial | 95.6 | .53 |
| 37 | **ดาลัด** — ขิบ | Crystal, glass | Malay | Literary | 69.6 | .39 |
| 38 | อับ — **ไพรสัณฑ์** | Forest | Sanskrit (*vana-saṇḍ*) | Literary | 97.0 | .57 |
| 39 | แผท — **ภคินี** | Sister | Sanskrit (*bhaginī*) | Royal | 98.2 | .53 |
| 40 | **เภตรา** — ขวัง | Ship, vessel | Sanskrit | Literary | 90.3 | .63 |
| 41 | วิมาส — **ธำมรงค์** | Ring | Khmer | Royal | 94.1 | .69 |
| 42 | เวรง — **วิฬาร์** | Cat | Sanskrit (*biḍāla*) | Literary | 93.5 | .32 |

The correct word in each pair is indicated in bold.

**Table 3. Psychometric properties and correlates of the Thai Lexical Decision Task (42 items) score distributions from the online and interview samples.**

| | | Study1 (online) | Study 2 (interview) |
|---|---|---|---|
| N = | | 662 | 90 |
| KR-20 | | 0.81 | 0.88 |
| Mean | | 38.32 | 32.58 |
| Median | | 39.00 | 34.00 |
| Range | | 15-42 | 17-42 |
| Coefficient of variance | | 10% | 21% |
| Correlation | Education level | 0.18 [a] | 0.42 [a, ***] |
| | Proficiency | 0.20 [a, ***] | 0.46 [a, ***] |
| | Age | 0.28 [a, ***] | 0.39[***] |
| | Gender | 0.09 [a, *] | .011 |

\*=$p$<.050, \*\*\*=$p$<.001, [a]=Spearman's RHO, proficiency refers to self-rated ability in Thai language.

further question addressed in the next study is whether scores on the Thai Lexical Decision Task are associated with performance on more-established vocabulary assessments.

## Study 2: Task validity with an interviewed sample

### Method

**Aims, study design, and hypotheses.** The aims of this study were to examine the validity and psychometric properties of the 42-item Thai Lexical Decision Task in greater detail, and in a more-educationally diverse sample. However, as this was one-to-one data collection, a paper copy of the test was provided, and participants asked to circle the real word in each word pair. This study was started simultaneously with the online study, but continued for a longer period, and so the full initial 100-item task was employed, the same set as in the online study. However, the analysis is limited to scores from the final 42-item version. To assess concurrent validity, a new sample of participants was recruited and assessed one-to-one with a range of Thai-language based measures. Validation was attempted by cross-sectional analysis of the correlations between the Thai Lexical Decision Task and other existing tests of Thai vocabulary. It was thus hypothesized that there would be sizable positive correlations between Thai Lexical Decision Task scores and other measures of vocabulary. It was also hypothesized that, as a measure or acquired crystallized ability, the positive associations found in the online study (Study 1) of task performance with age, education, and proficiency would be replicated.

**Participants.** A sample of 90 adult participants was recruited through advertisements on social media and personal contacts. The recruitment period was the same as Study 1. The mean age of the sample was 36.0 ($SD$=16.7, range 18–68). The majority were women ($n$=62, 69%), with 24 men ($n$=24, 27%), with the remainder identifying as other genders ($n$=4, 4%). The sample was educationally diverse, with the total years of formal education completed ranging from 4 to 29; however, the mean was still rather high at 17.6 years ($SD$=3.4). In terms of education level completed, 36/90 (40%) had high school or less as their highest completed level (compare this with the 4% reported in Study 1, described above). Indeed, the current (interview) sample had a statistically significant lower education level than the Study 1 (online) sample, $X^2$=165.75, $p$<.001, Cramer's $V$=.469. The level of 40% of the sample with high school or less as their highest education level is similar to the national level, which is 50% [48].

Although Thai is the official and most common language spoken in Thailand, the country is multicultural and other languages are frequently spoken, including English, Chinese, and indigenous tribal languages. Nevertheless, understanding

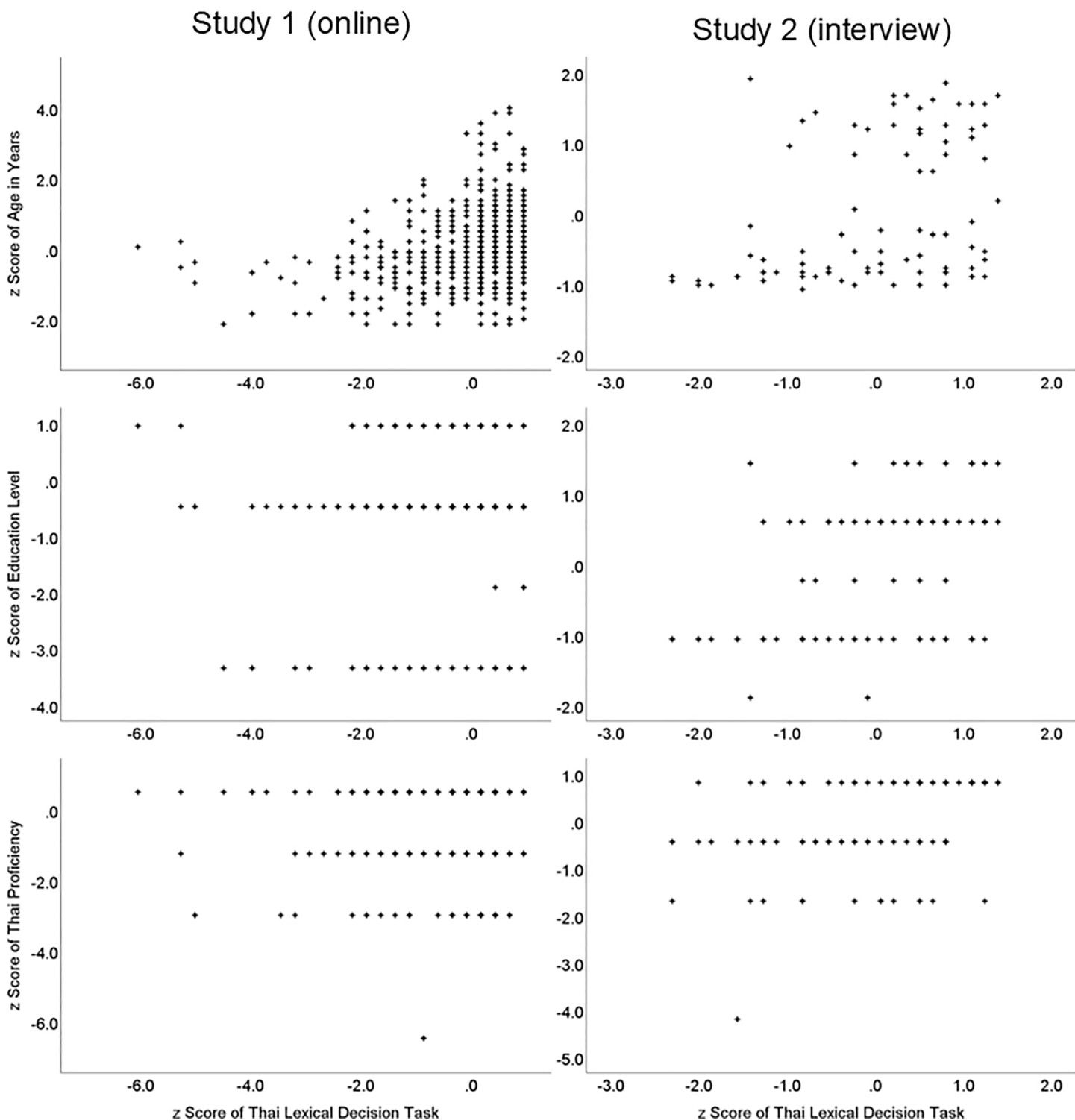

**Fig 1. Scatterplots of scores on the Thai Lexical Decision Task with demographic variables.**

Thai language was a study inclusion criterion. During the assessment, all participants were asked to rate themselves on proficiency in Thai language on a 5-point scale from very low to very high and asked about their exposure to Thai before the age of 5. The majority of the participants ($n$ = 89, 99%) were exposed to Thai language before the age of 5. The one exception nevertheless rated themselves as very high proficiency as an adult. In contrast, one participant rated their proficiency in Thai language as very low (a bilingual Thai and Chinese participant). The majority, though, rated their skill in Thai-language as 'high' (34%) or 'very high' (51%).

**Assessments.** To examine concurrent validity of the Thai Lexical Decision Task, we sought out established crystalized-cognitive measures of Thai vocabulary for adults. However, few such tests exist that could be considered already validated for use as criterion variables. The tests included were the best available options.

The main measure employed were verbal fluency tasks. In such tasks, participants are required to say as many words as possible from a specified set, within a time limit of one minute. Although the verbal output of the participant is oral, the words are written down by the test administrator. Various fluency tasks have been validated within the Thai population, mainly through neuropsychological studies. This is because they have been used as individual tests with patients with known cognitive impairments (e.g., [49]) or they have been used in computerized assessment studies (e.g., [50]), or they have been included as subtests in screening tools for cognitive impairments (e.g., [51]).

Verbal fluency tasks tend to measure both fluid (e.g., executive) and crystalized (i.e., language) ability [14]. Nevertheless, these tests are useful as criterion measures as they have been included in so many cognitive screens validated in Thailand that the soundness of their psychometrics properties is established. In addition, in healthy participants, as in the current study, performance appears to be limited mainly by variation in language knowledge, rather than executive function [52].

Several forms of verbal fluency task are available in Thai, differing on the set specified to the participant. We used six such tasks, one trial for each type. One type used was category fluency. This requires participants to produce words constrained by a semantic category. The categories we used were all included in previous studies in Thailand 1) Animals [49,51,53–55], 2) Fruits [49], and 3) First Names [55,56].

We also included phonemic fluency trials. In such tasks, the limiting factor given to the participants is that words should begin with a particular phoneme. We used fluency for ส,ก, and อ characters of the Thai writing script. These Thai characters have been used in phonemic fluency tasks in numerous validity and task-development studies [49–51,53].

We recorded one minute of responses for each of the six trials but distinguished within the data between the number of words reported in the first 30 and subsequent 30 seconds. This was achieved simply by having the researcher draw a line across the page at the 30 second point. We did this because it is thought that relatively automatic lexical memory searching is the principal skill indexed by performance in the first 30 seconds, later performance indexes more executive cognitive control, e.g., search strategy [57,58]. As we expected our lexical-decision task to assess breath of lexical stores (not executive control), we additionally hypothesized that correlations should be higher for scores from the first 30 seconds, than the subsequent 30 seconds.

We also used the Boston Naming Test, which is part of the Boston Diagnostic Aphasia Examination and is thus designed for neuropsychological use [59]. It involves patients being presented with drawings of objects and they are asked to name them. Although this is a clinical tool, a validated short form is available in Thai, and even healthy controls produced a range of scores without a substantial ceiling effect, suggesting that the test potentially measures vocabulary in non-clinical populations [60]. That version contains images of five low-frequency, and five medium-frequency objects. The total score range is 0 (worst) to 10 (best).

**Procedure.** The procedures applied, including taking written consent, were approved by the local ethics committee. All participants were recruited through advertisements or personal contacts, and provided written, informed consent. All were assessed face-to-face in a one-on-one interview setting.

## Results and discussion of the interview study

**Psychometric properties of the total score.** The psychometric properties of the 42-item Thai Lexical Decision Task in the 90 participants interviewed face-to-face are shown in Table 3, alongside the equivalent statistics from Study 1 (the data distribution can be viewed in S1 Fig). As in the parallel online study (Study 1), the scale had good internal consistency as estimated with KR-20. In contrast to Study 1 (online), in the interview sample the data distribution appeared to be normal, with skewness ($z = 2.40$) and kurtosis ($z = 1.15$) both within limits for a normal distribution [45]. Normality of data distribution was confirmed by examination of the Q-Q plot. Visual analysis of the data distribution did suggest that it was somewhat negatively skewed, but considerably less than in the previous online study. This is also reflected in the higher degree of variation in task performance scores in the current sample, compared to the Study 1 sample, based on the much higher coefficient of variance. This may be because of the increased diversity of educational background in the interview sample; indeed, the measures of central tendency for Thai Lexical Decision Task scores are around 5 points lower, reducing the previously observed ceiling effect. This may also be why the association between task scores and education was of a larger magnitude in this study, with a qualitative 'large' effect [46,47]. Similarly, there was a large positive association with age, likely also linked to the greater age diversity in this sample. Of the 20 high-frequency words included, the mean accuracy was 15.77 (SD = 3.34, 79%), which is somewhat higher than the score for the 22 low-frequency words in which the mean score was 16.81 (SD = 3.94, 76%). Nevertheless, the difference was not statistically significant, $t(89) = 1.79$, $p = .076$ (two-sided), $d = 0.189$.

The coefficient of variance was relatively large at 21%, which would indicate that the scale provides a reasonable amount of differentiation of scores across participants for it to be used as a measure of individual differences in performance. In fact, this is a higher value that reported for the English-language Spot-the-Word Test in non-clinical samples which had a coefficient of only 8% in a New Zealand study [61] and 9% in a USA-based study [62], and 16% in a UK-based study [63]. This may suggest that the scale could function well as a tool to measure individual differences, in which the ability to rank participants is an important feature of task reliability [64].

Bivariate correlations were performed to identify variables that may need to be controlled in later analyses. All cognitive tests scores were correlated with age of the participants, which is understandable considering the age-diversity of the sample (ages 18–68). Age effects on cognition are variable in adulthood, with more-fluid abilities tending to decline with age and more-crystallized abilities tending to increase with age [5]. Our cognitive tests scores showed the expected pattern, with lexical decision scores, the presumptive crystalized measure, showing a positive association, and the other tasks, thought to contain fluid processing aspects, tending to show negative associations. To negate these potentially confounding effects, age-adjusted data distributions were created for all cognitive measures by regressing test scores with age and storing the residuals. These age-adjusted scores were then used in later analyses.

**Correlations of Thai Lexical Decision Task with criterion variables.** The data distributions of the age-adjusted scores for the Thai Lexical Decision Task and all measures of verbal fluency, and the Boston Naming Test did not appear to vary from the normal distribution, based on analysis of skew and kurtosis nor visual inspection of distributions and Q-Q plots. Therefore, parametric correlations were used to assess the associations. These are shown in Table 4, and key scatterplots are shown in Fig 2.

**Table 4. Mean verbal fluency scores and Pearson correlation r values for the associations with Thai Lexical Decision Task scores.**

|  | Semantic | | Phonemic | |
| --- | --- | --- | --- | --- |
|  | Mean (*SD*) | r | Mean (*SD*) | r |
| 00-30 Sec. | 39.2 (9.4) | .51*** | 25.8 (8.2) | .40*** |
| 31-60 Secs. | 23.1 (5.6) | .38*** | 16.3 (6.8) | .34*** |
| 00-60 Secs. | 62.3 (13.4) | .51*** | 42.1 (13.4) | .42*** |

Means show sum of words over 3 trials, *** $p < .001$.

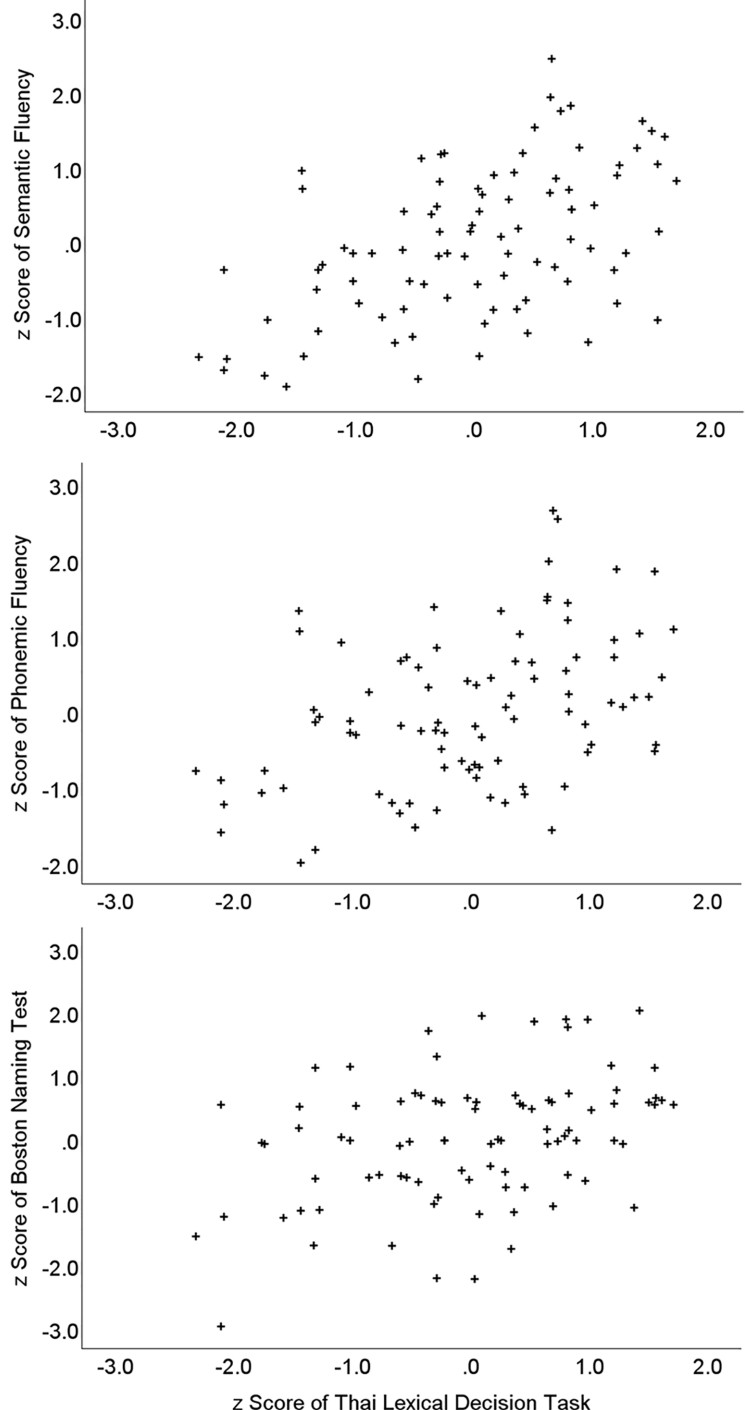

**Fig 2. Scatterplots of Thai Lexical Decision Task scores and other measures of language skill in the interview sample (Study 2).**

The pattern of correlations indicates large positive correlations between the Thai Lexical Decision Task and both semantic and phonemic fluency measures. The effect is strongest in the first 30-second period of the task, compared to the second 30-second period. This is consistent with past arguments that the first period of the task is more associated with automated searching of lexical entries, compared to the subsequent period of task performance in which performance variation is due to executive control factors [57,58]. Thus, of the correlation r values reported, those for the 0–30 second period are more relevant for validation purposes, adding further support to the validity of the Thai Lexical Decision Task as a measure of vocabulary. Not shown in the table is the large positive correlation between the Thai Lexical Decision Task and total verbal fluency performance (combining all fluency tasks trials), $r = .51$, $p < .001$.

We also examined the association of the Thai Lexical Decision Task with scores on the abbreviated Boston Naming Test. There was again a positive correlation between performance on the two tasks, $r = .38$, $p < .001$. Also relevant to assessing validity is the large correlation between Thai Lexical Decision Task scores and self-rated proficiency in the Thai language ($r = .46$, $p < .001$).

To interpret the magnitude of these correlations, for validity studies of cognitive tests, correlation coefficients can be qualitatively interpreted as being 'very high' if $r > .5$, 'high' if the r is between .40 and .49, and 'acceptable to moderate' if r values are from .20 to .39 [65]. The validity estimates in this study can therefore all be considered to range from moderate to very high, depending on the measures.

**Summary of the interview study.** To summarize the results from this study, the reduction from 100 items in the original version to the smaller 42-item Thai Lexical Decision Task appears to have produced a scale with good internal consistency. This therefore replicates the finding from Study 1 of good internal consistency in an independent sample. In addition, the scale produces a wide range of scores indicating its potential as an individual differences measure of ability. Furthermore, it can be seen as a validated measure of vocabulary, based on its associations with existing measures of vocabulary in Thai.

The final psychometric issue is whether the Thai Lexical Decision Task can be considered a reliable measure of vocabulary. As a self-completion scale, there is no issue with inter-rater reliability to address. Although split-half reliability could be calculated, that is a rather weak estimate of reliability as it is dependent on the ordering of items. Besides which, the KR-20 analysis already described indicated good internal consistency, and that is equivalent to the mean of all possible spilt-half reliability coefficients [44]. The remaining question is whether the Thai Lexical Decision Task has temporal stability, that is, a practice effect and test-retest reliability. This is examined in Study 3.

## Study 3: Temporal stability of the Thai Lexical Decision Task

### Method

**Aims, study design and hypotheses.** The aim of this study was to assess the stability of Thai Lexical Decision Task scores over time. To this end, a prospective follow-up of a subsample of participants from Study 2, described above, was performed. This allowed us to examine the practice effect and test-retest reliability. It was hypothesized that there would be a correlation ($r > .5$) between sessions when participants perform the Thai Lexical Decision Task on two separate occasions.

**Participants.** A subsample of 27 participants from the study described in Study 2 was retested. These were participants who indicated willingness to be assessed a second time, the sample size was based on that needed to detect r values $> .5$ at 80% power [66]. The demographics from this subsample did not differ significantly from the remainder of the sample described in that study (mean age = 37.5, $SD$ = 18.3; mean years of education = 18.4, $SD$ = 3.7; 67% women).

**Procedure.** The same procedure as for the previous study was employed, with the exception that neither the consent procedure nor collection of demographic information were repeated. Otherwise, the same task sequence and task rules were applied. Each participant was retested in the same location as the original data location reported in the previous study, with a small number of exceptions. All were assessed by the same research assistant who performed the initial

interview. Retest sessions were conducted approximately 5 weeks after the first test session (mean gap in days = 35, $SD$ = 10, range = 21–61).

**Assessment tools.**  The same assessment tools as described in the previous study were used again in this study.

### Results and discussion of the temporal stability study

There appeared to be no obvious practice effect, the mean score at the original test session was 32.2 ($SD$ = 7.2) and at retest it was 31.9 ($SD$ = 7.4), in fact, retest scores were marginally lower than test scores. To examine the test-retest reliability, we used Pearson correlation. This indicated a very large association between scores at the initial test session and the follow-up retest session, $r$ = .92, $p$ < .001 (Fig 3). That value exceeds even the most stringent criteria suggested for 'good' test-retest reliability [67].

**Summary of the temporal stability study.**  To summarize this study, scores on the Thai Lexical Decision Task appear to be very stable over time, indicating a lack of practice effect and high test-retest reliability.

### General Discussion

The overall aim of this research was to develop a standardized lexical-decision task for use with Thai-speaking participants. As a measure of acquired, crystalized vocabulary, the task showed the expected associations with self-report variables, being positively correlated with age, education level, and Thai-language proficiency. It also appears to be a psychometrically sound, valid and reliable measure of vocabulary. The task was designed for use in psychological research, and contains several quite low-frequency items. Although it could potentially be explored for use as a measure of Thai as a second language ability, it may be too difficult for less than advanced learners. Here we discuss in more detail some issues raised by the research.

### Critical evaluation of the methods and results

Initial data was collected using online methods. This has pros and cons. Although this allowed us to collect data on a relatively large sample of almost one thousand participants, the selection of participants was uncontrolled. To enhance data quality, a full third of the sample had to be excluded. Nevertheless, this still left a sizable sample which allowed us to use factor analysis methods to select a set of the best performing items loading onto a single latent factor, presumably lexical knowledge. An additional issue with data collection online using Qualtrics was the limited accuracy of response time

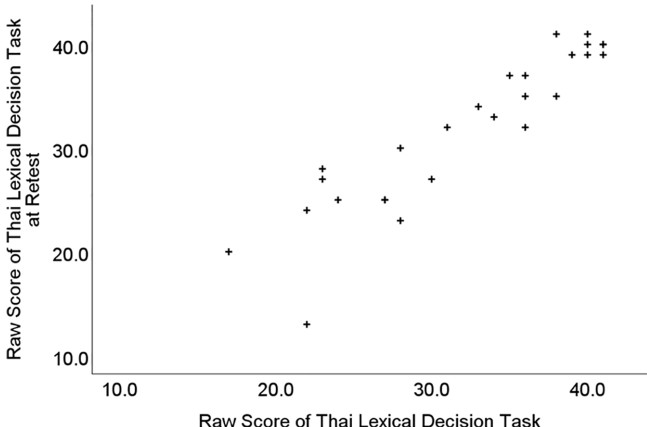

**Fig 3.  Scatterplot of Thai Lexical Decision Task raw scores from the 27 participants in Study 3 who also completed a retest assessment.**

data. However, the only use made of times was for excluding participants who completed the full task unusually quickly or slowly, thus no fine timing at the sub second level was required.

The eventual tool produced, the Thai Lexical Decision Task, has 42 items, each a real word paired with a pseudoword. This is somewhat shorter than similar tests available in other languages. For example, the original English-language Spot-the-Word test had 60 word-pairs [25,27] and a later version had 100 pairs [26]. Thus, the current test will produce a narrower range of scores than those tests. However, doubt has been raised as to whether so many items are even needed for the Spot-the-Word test to function as an estimator of crystalized intelligence (its principal use). Indeed, van der Linde et al. [63] have provided evidence that a reduced set of only 34 items performs better than the full set or 100 items for predicting full-scale IQ scores. We noted that there were only negligible word frequency effects on performance. Usually, word frequency has a large influence of lexical-decision tasks. However, it has been argued that the effect is actually driven by the number of contexts in which words are experienced [68]. This factor was not measured in the current study and may be a reason for the lack of a word frequency effect on performance. The absence of a word frequency effect also likely reflects the stimulus design, which relied on a restricted frequency range and predominantly low-frequency literary words that are not part of everyday language use.

## Potential applications

One reason that prediction of IQ scores by vocabulary tests, including lexical-decision tasks, is valued, is so that they can potentially be used as estimators of premorbid cognitive function for neuropsychological assessment. The tests used are usually either word pronunciation tasks in which phoneme-grapheme conversion would be ineffective (e.g., irregular spelled words in English), or lexical decision tasks (such as the Spot-the-Word Test). Often very high correlations are observed between such lexical tasks and full-scale IQ in healthy samples [6,35]. Because of this observation, regression formulas can be produced that can be used to predict IQ scores, or else the lexical test can be co-normed with an IQ test (e.g., [69]). Just on that basis, the lexical task can then rapidly estimate IQ scores. In fact, lexical tasks can be combined with demographic factors to produce regression-based test norms for various measures of cognitive ability, even when the correlations are not so strong [70].

However, to be effective as estimators of premorbid ability, performance of the lexical task has to be robust in the presence of disorders that would be expected to negatively impact cognitive function. This is based on the previously discussed distinction between fluid cognitive processes which are generally susceptible to cognitive impairment, and crystalized ability which is far less likely to be impaired in the presence of neurological or psychiatric disorders. Lexical decision tasks are designed to measure that crystalized ability [27] and evidence suggests that lexical-decision tasks are robust and 'hold', even in the presence of mild dementia. This has been shown with tests in various languages including English [71], Portuguese [34], and Spanish [6].

The current test, the Thai Lexical Decision Task, could potentially be used in this role, although further research would be needed to confirm that it too, as a measure of crystalized ability, is robust in the presence of cognitive impairment that nevertheless impairs fluid ability. Though, it seems likely that it would be. Firstly, there is no obvious reason why lexical decision tasks in Thai would not show such robustness when similar tests in other languages have been shown to be so. Secondly, we deliberately designed the Thai Lexical Decision Task to include items that would likely hold, as this would enhance them as indices of crystalized ability. We did this by using only concrete, imageable words. This was done because imageability is an important factor in whether word forms are susceptible to retrograde loss in dementia [72]. This is not a feature of tests in other languages such as the Spot-the-Word Test [25,27] or Spanish Lexical Decision Task [33].

It would also be useful to establish the strength of correlation with other cognitive tests. IQ is often used for premorbid estimations because vocabulary is extremely closely linked to general intelligence and may be the best single predictor of general cognitive ability [11,73]. This is a very useful association; because of the general manifold of positive test inter-correlations, knowing an individual's scores on a strong measure of general ability, particularly lexical measures [74], will allow some level of prediction of their performance on any cognitive test.

Nevertheless, such lexical tests generally have their strongest statistical associations with cognitive tests that have high verbal content [6,75]. As an example, the Spot-the-Word Test was originally intended as a premorbid predictor of speed of semantic processing of reading sentences [27], based on the strong correlation between the two different measures of around $r = .57$.

## Strength of validity evidence

At this point, it is worth considering the strength of the correlations reported in the current study between Thai Lexical Decision Task scores and measures of verbal fluency and picture naming (Boston Naming Test). Here, we were primarily interested in the use of correlation coefficients to establish test validity. The highest r values reported in this respect were a little over 0.5. Some authors have argued that values should be higher than that in validity studies, while others have suggested much lower thresholds, for example, that values over 0.5 could be considered 'very high' for purposes of validation [65,76]. In practice, the threshold used should reflect the similarity of the measures, the specific hypotheses made, etc. We feel that the current findings are indicative of the test's validity because the verbal fluency tasks and the Boston Naming Test were both rather different in format to the lexical decision task. As such, very high correlations could not be reasonably anticipated.

Furthermore, the relative sizes of the correlation coefficients for verbal fluency were consistent with our hypothesis that performance in the first half of the tasks would be a better indicator of lexical knowledge than performance in the second half. In fact, our results provide additional support to previous suggestions that the first 30 seconds of performance of verbal fluency tasks are more dependent on breadth of the lexicon, compared to the subsequent time spent on individual trials [57,58].

Given the simplicity of the Thai Lexical Decision Task, and its minimal involvement of cognitive control for completion, it appears to have substantial face validity. We therefore argue that the validity of the task in question as a measure of crystalized vocabulary knowledge is practically established.

We would also note that the values observed are in fact slightly higher than those typically found between another widely used measure of crystalized ability in English, the National Adult Reading Test [7], and the same tests used in this study, namely the verbal fluency and the Boston Naming Test [75]. Another potential application of the Thai Lexical Decision Task is as a brief estimator of verbal intelligence. Full intelligence tests can be very time consuming and impractical in research. Brief estimators of intelligence are therefore useful.

## Conclusion

We present a brief lexical decision task that can be used to assess crystalized vocabulary knowledge in Thai. The test appears to be an internally consistent, valid, and reliable measure of such knowledge. The Thai Lexical Decision Task is free to use in research without cost, and can be downloaded from: https://gpluck.co.uk/Tests/

## Supporting information

**S1 Fig. Distribution of scores on the 42-item Thai Lexical Decision Task for the 90 participants in Study 2 (interview).**
(DOCX)

## Acknowledgments

We would like to thank Nutvara Patitus, Kamorichan Osthananda, Panyada Khongthong, and Saranwat Jetsawangsri for assistance with data collection, and Fei Gu for assistance with conducting the factor analysis.

## Author contributions

**Conceptualization:** Graham Pluck, Suphasiree Chantavarin.

**Data curation:** Alexis Sirisomboonwong.

**Formal analysis:** Graham Pluck, Carl Piaf.

**Investigation:** Graham Pluck, Alexis Sirisomboonwong, Smriti Sitani, Suphasiree Chantavarin.

**Methodology:** Graham Pluck, Smriti Sitani, Carl Piaf, Suphasiree Chantavarin.

**Project administration:** Alexis Sirisomboonwong.

**Writing – original draft:** Graham Pluck.

**Writing – review & editing:** Alexis Sirisomboonwong, Smriti Sitani, Carl Piaf, Suphasiree Chantavarin.

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
