## [Decision Letter · Decision Letter 0]

25 Jun 2025

Dear Dr. Chantavarin,

We look forward to receiving your revised manuscript.

Kind regards,

Yiu-Kei Tsang

Academic Editor

PLOS ONE

Journal Requirements:

2. Please update your submission to use the PLOS LaTeX template. The template and more information on our requirements for LaTeX submissions can be found at http://journals.plos.org/plosone/s/latex....

Reviewers' comments:

Reviewer's Responses to Questions

**Comments to the Author**

1. Is the manuscript technically sound, and do the data support the conclusions?

Reviewer #1: Partly

Reviewer #2: Yes

2. Has the statistical analysis been performed appropriately and rigorously?

Reviewer #1: Yes

Reviewer #2: I Don't Know

3. Have the authors made all data underlying the findings in their manuscript fully available?

Reviewer #1: Yes

Reviewer #2: No

4. Is the manuscript presented in an intelligible fashion and written in standard English?

Reviewer #1: Yes

Reviewer #2: Yes

Reviewer #1: This manuscript addresses an important area of vocabulary assessment in a lesser-studied language, which is a valuable contribution to the field. The overall aim is clear, and the two-study design has the potential to offer meaningful insights into vocabulary knowledge measurement. However, there are several conceptual and methodological issues that need to be addressed to strengthen the clarity, theoretical grounding, and validity of the test. Clarifying the target population, refining the theoretical framework, and providing more transparent reporting of stimuli and procedures will significantly enhance the paper’s overall impact and interpretability.

1. Target Population and Terminology: The target population for the test is not clearly defined. Additionally, the manuscript inconsistently uses the terms “patients” and “participants,” which should be clarified.

2. Conceptual Framing of Vocabulary Assessment: The definition of vocabulary assessment is drawn from literature on healthy adults, but many cited examples relate to verbal intelligence rather than vocabulary breadth. To align with the stated aim of measuring vocabulary knowledge, it would be beneficial to review work on lexical decision and yes/no vocabulary tests, particularly Meara’s research and LexTALE-inspired instruments.

3. Theoretical Framework: The distinction between word recognition and recall is relevant and well-covered in Laufer and Aviad-Levitsky’s work, which could further strengthen the discussion. A clearer theoretical justification is needed to explain how the test format captures vocabulary knowledge, especially given that forced-choice tasks may reflect perceived word-likeness rather than lexical knowledge per se.

4. Stimuli Transparency: Providing English translations for the Thai stimuli would enhance accessibility for readers unfamiliar with the Thai language.

5. Item Characteristics: The paper should report the proportion and characteristics of low- vs. high-frequency words. This information is crucial to understand the lexical difficulty of the test.

6. Outcome Measures and Platform Limitations: The outcome measures for the lexical decision task are not clearly described. Additionally, Qualtrics may not be suitable for collecting accurate response times, and this limitation should be acknowledged.

7. Methodological Justification: Selection criteria, data cleaning procedures, and item analysis should be supported by relevant literature to enhance methodological rigor.

8. Interpretation of Test Format: It remains unclear whether the task truly measures vocabulary knowledge. The format, which asks participants to choose between a real word and a nonword, may primarily assess word-likeness judgments rather than vocabulary breadth. This should be addressed in the theoretical rationale.

9. Study 2 – Criterion Measures: The purpose and nature of the criterion tasks in Study 2 are unclear. In particular, the description of “verbal fluency tasks” lacks detail, making it difficult to determine whether they assess spoken or written fluency. For example, the description on p.19, line 391, suggests a written task, which needs clarification.

Useful references:

1. Vermeiren, H., Vandendaele, A., & Brysbaert, M. (2023). Validated tests for language research with university students whose native language is English: Tests of vocabulary, general knowledge, author recognition, and reading comprehension. Behavior research methods, 55(3), 1036–1068. https://doi.org/10.3758/s13428-022-01856-x

2. Lee, S. T., van Heuven, W. J. B., Price, J. M., & Leong, C. X. R. (2025). Assessing bilingual language proficiency with a yes/no vocabulary test: the role of form-meaning vocabulary knowledge. Bilingualism: Language and Cognition, 1–17. https://doi.org/10.1017/S1366728924001007

3. Aviad-Levitzky, T., Laufer, B., & Goldstein, Z. (2019). The New Computer Adaptive Test of Size and Strength (CATSS): Development and Validation. Language Assessment Quarterly, 16(3), 345–368. https://doi.org/10.1080/15434303.2019.1649409

4. Schmitt, N., Nation, P., & Kremmel, B. (2020). Moving the field of vocabulary assessment forward: The need for more rigorous test development and validation. Language Teaching, 53(1), 109–120. https://doi.org/10.1017/S0261444819000326

Reviewer #2: General Comments:

This is a well-written and well-argued manuscript.

1. Introduction: The authors present the lexical decision task as a tool to identify the real word among a pair or group of letter strings and note the lack of standardised versions for clinical or research use. While this may apply to psychological research, such lexical decision tasks are also used in language sciences—for instance, the Dialang and LexTALE tests. The French LexTALE is even mentioned in the article, but not the English one. However, the task is different (yes/no recognition task). Some second language acquisition scholars suggest that these tasks are good proxies for general language competence. I suggest expanding the literature review to reflect this broader context or explaining why it does not need to be included.

2. Empirical Studies:

I recommend summarising key participant characteristics in (a) table(s) for clarity. In the results section, including plots/graphs is necessary to help readers better visualise data distribution and relationships.

Some clarifications are requested directly in the annotated version of the article. More detailed feedback is provided in this attached Word document.

.

Reviewer #1: No

Reviewer #2: No

---

## [Author Response · Author response to Decision Letter 1]

25 Dec 2025

We have addressed all of the comments from the reviewers, as outlined in the Response to Reviewers file. Additionally, we have also added data repository information to the manuscript. We have also provided a PDF file of the manuscript in LaTeX format.

---

## [Decision Letter · Decision Letter 1]

26 Feb 2026

We look forward to receiving your revised manuscript.

Kind regards,

Yiu-Kei Tsang

Academic Editor

PLOS One

Journal Requirements:

Reviewers' comments:

Reviewer's Responses to Questions

**Comments to the Author**

Reviewer #1: All comments have been addressed

Reviewer #3: All comments have been addressed

2. Is the manuscript technically sound, and do the data support the conclusions?

Reviewer #1: Partly

Reviewer #3: Yes

3. Has the statistical analysis been performed appropriately and rigorously?

Reviewer #1: Yes

Reviewer #3: I Don't Know

4. Have the authors made all data underlying the findings in their manuscript fully available?

Reviewer #1: Yes

Reviewer #3: Yes

5. Is the manuscript presented in an intelligible fashion and written in standard English?

Reviewer #1: Yes

Reviewer #3: Yes

Reviewer #1: Most of my previous comments were addressed by the authors. A few additional notes:

Pg. 5 - line 104 - typo "let"

Pg. 17 - line 346 - It's confusing to me here. correlation between .1 - .3 are generally considered weak, .3-.5 are considered moderate, and >.5 are considered strong. It's unclear to me why .3 and above are considered "large".

Reviewer #3: This paper reports a lexical decision task that can potentially be used to assess crystalized vocabulary knowledge in Thai. The authors have responded clearly to the two reviewers’ comments and have clarified important issues raised earlier.

It would be beneficial if the authors could describe the selection criteria as well as the characteristics of the real words included in the list in greater detail. For example, some words belong to royal vocabulary; some have Pali or Sanskrit roots, while others are native Thai words; some are found only in poetry or traditional literature, etc. (As a note, ชลมารค does not mean ‘royal barge’; it should be translated as ‘watercourse’ or ‘waterway.’) The process of pseudoword formation could also be elaborated. Some pseudowords appear to be modified real words, while others seem to be combinations of parts of real words. Providing this kind of detail would help readers who are not familiar with Thai gain a clearer picture of the nature of the words and pseudowords used in the study.

On p. 14, the authors state that 20 of the 42 words were relatively high-frequency items, while the remaining words were relatively low-frequency. Did frequency have an effect on participants’ performance?

.

Reviewer #1: No

Reviewer #3: No

---

## [Author Response · Author response to Decision Letter 2]

9 Apr 2026

Please see our detailed responses in the Response to the Reviewers file.

---

## [Editor Report · Decision Letter 2]

12 Apr 2026

A lexical decision task for rapid estimation of crystalized vocabulary knowledge in Thai

PONE-D-25-14461R2

Dear Dr. Chantavarin,

We’re pleased to inform you that your manuscript has been judged scientifically suitable for publication and will be formally accepted for publication once it meets all outstanding technical requirements.

Kind regards,

Yiu-Kei Tsang

Academic Editor

PLOS One
---

## [Editor Report · Acceptance letter]

PONE-D-25-14461R2

PLOS One

Dear Dr. Chantavarin,

I'm pleased to inform you that your manuscript has been deemed suitable for publication in PLOS One. Congratulations! Your manuscript is now being handed over to our production team.

Kind regards,

on behalf of

Dr. Yiu-Kei Tsang

Academic Editor

PLOS One